# Impact of Magnetic-Pulse and Chemical-Thermal Treatment on Alloyed Steels' Surface Layer

**Kateryna Kostyk** [1], **Ivan Kuric** [2], **Milan Saga** [3], **Viktoriia Kostyk** [4], **Vitalii Ivanov** [5,*], **Viktor Kovalov** [4] **and Ivan Pavlenko** [6]

[1] Department of Foundry, Educational and Scientific Institute of Mechanical Engineering and Transport, National Technical University "Kharkiv Polytechnic Institute", 2, Kyrpychova, 61002 Kharkiv, Ukraine; Kateryna.Kostyk@khpi.edu.ua

[2] Department of Automation and Production Systems, Faculty of Mechanical Engineering, University of Zilina, 8215/1, Univerzitna St., 010 26 Zilina, Slovakia; ivan.kuric@fstroj.uniza.sk

[3] Department of Applied Mechanics, Faculty of Mechanical Engineering, University of Zilina, 8215/1, Univerzitna St., 010 26 Zilina, Slovakia; milan.saga@fstroj.uniza.sk

[4] Department of Computerized Mechatronic Systems, Tools and Technologies, Faculty of Machine Building, Donbas State Engineering Academy, 72, Akademichna St., 84313 Kramatorsk, Ukraine; vikakostik777@gmail.com (V.K.); kovalov.viktor@gmail.com (V.K.)

[5] Department of Manufacturing Engineering, Machines and Tools, Faculty of Technical Systems and Energy Efficient Technologies, Sumy State University, 2, Rymskogo-Korsakova St., 40007 Sumy, Ukraine

[6] Department of Computational Mechanics Named after V. Martsynkovskyi, Faculty of Technical Systems and Energy Efficient Technologies, Sumy State University, 2, Rymskogo-Korsakova St., 40007 Sumy, Ukraine; i.pavlenko@omdm.sumdu.edu.ua

* Correspondence: ivanov@tmvi.sumdu.edu.ua; Tel.: +380-664880319

**Abstract:** The relevant problem is searching for up-to-date methods to improve tools and machine parts' performance due to the hardening of surface layers. This article shows that, after the magnetic-pulse treatment of bearing steel Cr15, its surface microhardness was increased by 40–50% compared to baseline. In this case, the depth of the hardened layer was 0.08–0.1 mm. The magnetic-pulse processing of hard alloys reduces the coefficient of microhardness variation from 0.13 to 0.06. A decrease in the coefficient of variation of wear resistance from 0.48 to 0.27 indicates the increased stability of physical and mechanical properties. The nitriding of alloy steels was accelerated 10-fold that of traditional gas upon receipt of the hardened layer depth of 0.3–0.5 mm. As a result, the surface hardness was increased to 12.7 GPa. Boriding in the nano-dispersed powder was accelerated 2–3-fold compared to existing technologies while ensuring surface hardness up to 21–23 GPa with a boride layer thickness of up to 0.073 mm. Experimental data showed that the cutting tool equipped with inserts from WC92Co8 and WC79TiC15 has a resistance relative to the untreated WC92Co8 higher by 183% and WC85TiC6Co9—than 200%. Depending on alloy steel, nitriding allowed us to raise wear resistance by 120–177%, boriding—by 180–340%, and magneto-pulse treatment—by more than 183–200%.

**Keywords:** sustainable manufacturing; alloy steel; hard alloys; surface hardening; magnetic-pulse treatment; nitriding; boriding; layer thickness; hardness

## 1. Introduction

Most parts and tools work on high-wear surfaces. Therefore, there is a need to protect the surface during operation. Improving wear resistance for cutting tools and machine parts made from traditional tool materials effectively increases performance. The surface layers' wear resistance for steel parts can be improved using various surface-hardening treatment methods. The material's physical-mechanical, thermophysical, and crystal-chemical properties strongly affect the cutting tools and machine parts. The optimal

combination of these properties allows us to control the wear, transform the wear mode, and reduce the product's contact pads' wear rate.

The diversity of surface hardening treatment methods is classified into the following groups: deformation effects, thermal stresses, surface alloying, coatings, and combined treatment [1]. An increase in the reliability of machines and tools is an urgent problem. It is complicated by metal parts' operating modes supplemented by friction, high thermal and mechanical loads, and aggressive environments.

Many surface hardening methods are based on coating or modifying the surface condition. Traditional approaches for increasing the efficiency of steel parts do not adequately provide the required properties. An essential objective is to increase the lifetime and wear resistance of machine parts and tools by reinforcement. This problem's solution allows us to significantly increase their longevity and save expensive and scarce materials, energy, and labor.

Existing chemical-heat treatment methods typically provide performance parts under friction and wear conditions. However, these methods are rather long and require comprehensive and expensive equipment [2].

Thus, the actual issue is the search for new methods to improve the performance of tools and machine parts. The solution to this problem should increase the hardening of surface layers and develop methods that significantly increase the hardening process without the deterioration of the products' properties. Methods belonging to different groups impact the surface layer of cutting tools and machine parts differently.

The material's surface layer is hardened when the deformation effect [3] occurs. In this case, its microgeometry and energy supply are changed. As a result, a thermal effect on the surface layer of alloys changes the layer's structure. However, its chemical composition remains unchanged.

Surface alloys change the surface layer's structure and chemical composition. The formation of a thin film supplements coatings on the tool's surface and parts. The surface hardening methods lead to hardening the product's surface layer due to the following mechanisms and combinations: the substructure, solid solution, polycrystalline, and multiphase hardening [4].

The surface hardening methods based on deformation effects are not widely used in manufacturing cutting tools. They are mainly used for hardening a wide range of machine parts.

For making deformations, hardening includes blasting [5], magnetic-pulse processing [6], hardening by explosion [7], smoothing and rolling [8], and ultrasonic treatment [9]. Each hardening processing method based on the deformation effect on the cutting tool has the following characteristics: mechanics of deformation, specific features of formation of geometrical and physical-mechanical properties of the surface layer, and the process conditions.

Work hardening methods are divided into either static and dynamic ones. Static methods (e.g., rolling and pressing) are characterized by the stationarity of the deformation force and the impact and continuity of contact between the deforming element and the cutting tool. Dynamic methods (e.g., blasting, vibration, ultrasonic treatment, and embossing) are characterized by the deforming elements' intermittent contact and impulsive effect on the cutting tool's surface [10]. Additionally, widespread ways for combining static and dynamic loads are applied.

All methods of magnetic treatment of tools and machine parts are classified as follows [6]: the processing of static magnetic field (intensity 100–1000 kA/m for exposure 10–300 s); magnetic-pulse processing field (intensity 50–2000 kA/m at a pulse duration of 0.1–10 s).

The first method includes the following stages: processing demagnetization during 8–24 h; processing a single pulse with a directional (local) concentration of the magnetic flux on the workpiece; dynamic processing when the item is in a field of constant tension and rotates with a rotational speed of $1$–$50 \text{ s}^{-1}$ during 1–5 min; treatment with the free movement of the workpiece in the inductor's cavity.

The magnetic-pulse treatment method includes the following stages: treatment without subsequent demagnetization; polycyclic processing (2–10 cycles) with exposure between cycles 1–20 min; processing using ferromagnetic cores; the local magnetic field concentrators; and machining in metal containers or chambers using ferrofluid. Almost all of these methods need the subsequent exposure of the workpiece on non-metallic substrates during 5–24 h.

Tools made of carbon steel need high-speed processing by 1–2 cycles under the pulse duration of 0.3–1.5 s and exposure after treatment of approximately 8–12 h. Increasing the tool's mass increases the optimum pulse time 2–3 times.

Tools made of hard alloys and composite materials need processing by 5–10 cycles under intensity of up to 4000 kA/m in the container with ferromagnetic fluid. In this case, exposure between treatment cycles depends on the tool's mass, geometry, and dimensions, which varied in a range of 0.5–5 min [10].

The most promising direction to increase the resistance of cutting tools from ferromagnetic materials is the processing of magnetic-pulse treatment. This treatment allows us to increase tool life due to changes in the mechanical properties of materials. This effect is achieved due to the complex magnetostrictive structural changes.

For the surface hardening of tools and machine parts, the following procedures can be applied: thermal impact [11], laser hardening [12], electron-beam processing [13], cryogenic processing [14], flame quenching [15], and the quenching of high-frequency currents [16]. Among the various methods, surface alloying is the most applicable in producing tools from alloy steels after thermochemical treatment [17]. The chemical-heat treatment of cutting tools includes the diffusion saturation technology applied to the surface layer for elements C, N, B. In this case, cementation, nitriding [18], nitrocarburizing [19], and boriding [20] were used. Notably, the choice of a proper chemical-thermal treatment method depends on the tool surface layer requirements and heat resistance of the tool's material.

In ion implantation in the surface layer of the irradiated material, radiation defects occur [21]. These effects lead to changes in the material's properties (e.g., microhardness, toughness, ductility, thermal conductivity, and electrical resistance). However, ion implantation is not widely used in tool production due to the high cost and relatively low performance. In the case of implanting the layers with significant thickness, these disadvantages are especially evident.

Laser doping [22], electro-erosive alloying [23] and plasma doping [24] are also common. Laser surface hardening realized by highly concentrated radiation is focused on a small area (fractions in a range of 1–10 mm) [25]. This treatment method hardens tools including high-speed steel, hard alloys, and ceramics. However, the hardening of the material's surface layer by changing its structure and chemical composition is reached at a shallow depth (up to 80 μm). There are disadvantages of electron-beam processing: the need for protection from X-radiation at voltages above 20 kV; relatively high cost; relative complexity of the technological equipment.

One of the most effective ways for ensuring an optimal mix of "strength-ductility" cutting tool materials are the wear-resistant coatings methods, high-temperature techniques (HT), and chemical vapor deposition (CVD), physical vapor deposition (PVD), electrolytic method, gas-thermal spraying, and welding [26].

The CVD process is generally based on heterogeneous chemical reactions at the vapor–gas medium surrounding the cutting tools. As a result, a durable coating is obtained [27]. Coatings of refractory compounds by deposition from the gas phase are based on the recovery of volatile compounds of metals with hydrogen in the presence of the gas mixture's active components. These interact with the released components which are free of metal and form the corresponding refractory compound. The HT-CVD process is realized at relatively high temperatures (up to 1100 °C). Such temperatures exclude this method for tools made of high-speed steel. However, the application of coatings on carbide tools heated to such high temperatures often adversely affects their operation. Carbide tools with CVD-coatings have an increased tendency to brittle fractures and chipping. This fact is

particularly evident in interrupted cutting when cyclic loading is applied, e.g., in processing difficult-to-machine alloys with large shear layer thicknesses. The coating cannot resist the cyclic loads because the thickness usually does not exceed 3–5 μm (i.e., while turning, this value may reach 15 μm).

The main direction of improving hard alloy with CVD-coating is to reduce the surface layer's fragility. The MT-CVD technology is also unable to solve this problem. The magnitude of the tensile stress in the surface layer of the coatings is slightly lower than that obtained by a high-temperature CVD method. However, it is still sufficient to lead to nucleation cracks.

When using the PA-CVD coating on the tool, chemical reactions in the gas phase are formed when exposed to a plasma of gas past a high-frequency electric discharge [28]. Compared with the CVD, the main advantages of the PA-CVD process are the fact that there is no need to heat the tool based on high temperatures (up to 600 °C) and the high strength of the adhesive bond for the coating and tool's basis. Simultaneously, controlling the PA-CVD processes is often quite complicated. This method's deposition of pure materials is impossible since the sediment retains almost all required gases. Another disadvantage is the strong interaction of the plasma with the growing film. A high deposition rate leads to the poor control of the homogeneity and requires careful debugging of the reaction setup.

The methods considered above to improve the wear resistance of cutting tools and machine parts can be used as the combined hardening treatment. However, the production found the use of only some of them, such as wear-resistant coatings [29], laser processing [30], ion nitriding [31], wear-resistant coatings [32], laser alloying [33], nitriding [34], and cryogenic-erosion processing [35].

Thus, traditional methods for processing machine parts and tools have many outstanding issues. Mainly, these do not provide sufficient hardened layer thickness, require lengthy processes, are hard to use, and are energy-consuming.

Therefore, the permanent increase in processing characteristics for obtaining high-performance properties of products needs to find a promising direction in which to improve the efficiency of cutting tools and machine parts. This direction should be closely complemented by developing effective surface hardening technologies. Overall, developing methods that significantly reduce the hardening process without compromising the product's properties is an urgent problem.

This work aimed to study the impact of different modes of hardening treatment, such as magnetic-pulse treatment (MPT) and chemical-heat treatment (CHT), to modify the surface layer alloyed steels' properties and hard alloys.

For achieving this goal, the following objectives should be solved:

- Study the changes of surface hardness and the thickness of the hardened layer of the carbide cutting tool formed using a modified method of processing pulsed magnetic field (PMF);
- Determine the surface hardness and thickness of the hardened layer of alloy steels after nitriding in a nitrogen-containing nano-dispersed substance with activators;
- Investigate the hardened layer's changes in alloyed steel's surface hardness and thickness after boriding in the nano-dispersed powder.

## 2. Materials and Methods

As the research object, carbide inserts WC85TiC6Co9 AND WC92Co8, with a size of 15.875 × 15.875 × 4.760 mm, were used for turning the walk-through and boring cutters and end mills. Cylindrical samples with a diameter of 12 mm and length of 20 mm of alloyed steels, bearing steel, high-speed tool steels were used in the quantity of 10 pieces per alloy.

Magnetic pulse hardening was carried out on the installation "MIU SFT 9.120.00.00.000" (manufactured at the Institute of Physics and Technology of the Republic of Belarus). The installation for metals' magnetic-pulse treatment was a pulse current generator (PCG). This consisted of the capacitive storage of electrical energy high voltage capacitor banks,

inductors, and switching device's working body high-voltage controlled dischargers. Due to the arrester's help, the discharge of the capacitor bank to the inductor occurs [36].

The control device was used to control the process of magnetic pulse hardening. This includes a control circuit of the time relay "VL-159M" and actuators in relays and magnetic starters.

Device control allowed us to set the product's processing mode, given the specific pulse energy, by setting the charge time. The last one was realized using the programmable time relay VL-159M, which governs the product's processing. This approach allowed us to more accurately select the modes of processing products. The selection of the functional diagram's type and setting the desired exposure time was also realized using this time relay. According to the established modes, automatic ignition and discharge were controlled due to the processing of the uploaded items.

The processing of samples from bearing steel Cr15 was applied according to the following sequence: the installation of the samples in the inductor; the choice of treatment modes; the impact of a pulsed magnetic field on samples; the removal of the processed samples; and subsequent aging during 24 h. After processing, the samples' aging was required to complete the internal processes associated with electromagnetic energy scattering in the material [36]. Experiments were performed at an energy of 6.56 kJ with the number of pulses from 2 to 9.

Magnetic treatment of inserts of solid alloys was carried out on the robotic complex "PPMF RK-1" (manufactured at the Donbass State Machine-Building Academy of Ukraine) based on the processing by pulse magnetic field modes. This complex provides the most significant boost to this value (the field intensity—$1.1 \times 10^5$ A/m; the duration of the MPT—2 min; the exposure time after processing—28 h; pulse frequency—5 Hz). It consists of a pulse generator with a power supply and inductor. The pulse generator and power supply were designed as individual stand equipment. The inductor was connected to the generator cable [37].

The inductor was located on the horizontal dielectric diamagnetic surface (e.g., plastic, wood, rubber) to handle small products. Its axis should be vertical. Products were placed inside the inductor, and the processing session lasted 120 s. The main technological parameter of the control unit is the operating voltage of the unit as a discharge voltage of the capacitor circuit generating pulses of a magnetic field. The corresponding value was displayed using a particular device on the front panel of the generator unit. Regulation of the operating voltage was carried out by rotating the handle. The solenoids' geometry for magnetic inductors concerning their optimality and ability was analyzed to provide the required magnetic field intensity values. The weakening of the working gap's magnetic field increases the solenoid's length.

The strengthening treatment of tools and machine parts was considered in saturating the surface of steel samples by atomic elements—nitrogen and boron. Nitriding was carried out in the nanosized nitrogen-containing substances with activators [38,39]. Before nitriding, the steels samples were subjected to hardening with subsequent high tempering. The surfaces of the samples of steel 40Cr, bearing steel Cr15, high-speed tool steel W6Mo5, and high-speed tool steel W18 were previously cleaned and degreased from traces of scale, rust, oil, and other contaminants. Samples were placed in a container filled with a mixture. The container was hermetically closed and placed in a chamber furnace. Nitriding was carried out in the temperature range of 450–650 °C during 1–7 h.

The boriding of steels was performed after annealing. For completing the paste's drying in an oven, a layer of a paste from nano-dispersed powders with a thickness of 2–3 mm was applied to the prepared samples. Samples with a printed layer of paste were placed in a crucible and covered with the boron compound. Boriding was carried out in the temperature range of 800–1000 °C during 15–120 min.

Optical microscopy on the microscope "MIM-8" (Russia) by the standard method at different magnifications was studied by studying the microstructure and thickness of diffusion layers.

The depth of the nitrided layer was taken at a distance from the surface to the layer, which has a hardness different from the core's one by 50 MPa. The borated layer depth was measured from the surface of the core sample. Hardness tester "PMT-3" (Russia, Vickers method) at a load of 50 g and 100 g was used during 7–15 s to measure the microhardness of the used. Different loads were used to measure different layers. That is, to measure a harder borated layer, a load of 100 g was required (a load of 50 g gives too small an imprint). Therefore, for the accuracy of the microhardness readings for layers of different hardness, it is advisable to apply a different load. To simplify the recording and perception of the results, the microhardness values were converted to GPa; for example, 20 GPa corresponds to 2000 $HV_{100}$. It is the average values of microhardness that are indicated in the study. For each hardened (diffusion) layer, fifteen injections (measurements) of microhardness were made for each sample. Five samples were examined for each treatment mode. The measurement error was 5%.

For the investigation of steels and alloys, a survey of diffraction patterns was carried out by X-ray diffractometer general purpose "DRON-3M" (Russia). The shooting for steels was carried out in chromium X-ray radiation. Cu X-ray radiation was used for hard metals.

For wear tests, the abrasive wear resistance was investigated on the stand "AP 40.613.20 R 43/82" (Germany). The degree of wear was determined by monitoring the weight loss of the test specimen.

## 3. Results

### 3.1. Experimentally Determining the Thickness of the Hardened Layer and the Surface Hardness after MPT for Products of Bearing Steel Cr15

The average value of microhardness for samples before treatment was 2.4 GPa. The presented dependences analysis shows that the processed samples' microhardness increased by 40–50%. The depth of the hardened layer is 80–100 μm [40].

Within the new technology of magnetic pulse hardening, the magnetic-pulse effect of a substance changes its physical and mechanical properties. The interaction of a pulsed magnetic field with the conductive material's workpiece is effective at higher degrees of structural and energetic heterogeneity. Therefore, a higher surface and internal stress concentration in metal parts indicates a greater probability of a local concentration of the external field's microwires. This effect leads to heating areas around the crystals of tense blocks inhomogeneities of the metal structure. A higher thermal flux gradient with magnetic-pulse treatment indicates that the metal's structure is less homogeneous. After processing, the alloy's microstructure is improved, leading to a change in the material's physical and mechanical characteristics.

### 3.2. Experimentally Studying the Effects of MPT on the Reliability of Cutting Tools of Hard Alloy WC85TiC6Co9

The MPT impact on the stability of the cutting properties of inserts from hard alloys (e.g., WC85TiC6Co9) during rough turning was assessed. The change in microhardness before and after treatment was analyzed. Based on the analysis of microhardness distribution on the hard alloy surface, it was found that MPT increases on average from 16.1 GPa to 16.9 GPa. The variation coefficient of the microhardness decreases from 0.13 to 0.06.

At the initial state and after processing, X-ray analysis was used as the research method. It is the most effective method for determining the structural characteristics of multi-phase crystalline solids [19]. The change in lattice parameters of Co and TiC after treatment shows that the lines (100) Co and (220) TiC shift toward larger angles. This fact indicates a decrease in lattice parameter distortions, confirms the cobalt phase WC85TiC6Co9 carbide deformation, and indicates improved hard alloy strength [41].

When machining hard alloys, the influence of the PMF on the structure and properties of the components was studied. It is known that the binder phase largely determines the strength of hard alloys. Surveillance of the cracks showed that destructive cracks spread in alloys of WC92Co8 and the WC85TiC6Co9, mainly in the cobalt phase. In WC85TiC6Co9 alloys, the destructive crack mainly spreads by phase (Ti,W)C, the cobalt component can

inhibit the destructive crack. This is explained by the fact that the cobalt phase of hard alloys is a solid solution of W and C in cubic Co.

An orderly arrangement of atoms has lower internal energy than a disorderly arrangement, mainly if the atoms distribution in the crystal lattice occurs at relatively low temperature when the entropy associated with the disorder plays a less significant role.

Changes in the cobalt phase's properties after the MPT are reduced to atoms' rearrangement under a magnetic field's influence. As regards the energy required for the recrystallization or separation of the dispersed phase, a higher energy magnetic field creates the MPT effect. This treatment does not influence the phase composition and texture of the material. However, the selection phase, crystallization, or stress can develop in addition to crystallographic directions such that the energy of crystallization or separation will be minimal in a specific direction depending on the magnetic field direction. An increase in the hard alloy's cutting properties' stability is associated with cobalt phase homogenization under a pulsed magnetic field. The application of vibro-abrasive treatment before exposure to the magnetic field enhances the intensity of the transition stresses in the cobalt phase from tensile to compressive. This effect increases the strength and stability of the cutting properties of the tool.

The analysis of the character of changes in values of lines with different MPT modes showed that there is a value of magnetic field intensity in MPT for solid alloys. This field produces the maximum displacement of the cobalt phase's diffraction lines and increases the intensity of the line (100) Co, which equals approximately $1.8 \cdot 10^5$ A/m.

Finally, studies have shown that a cutting tool equipped with inserts from hard alloys WC85TiC6Co9 and WC92Co8 has a resistance relative to the untreated WC92Co8 higher by 183%, and WC85TiC6Co9—more than 200%. The process of wearing inserts from WC92Co8 and WC85TiC6Co9 is characterized by the absence of plastic deformation in the cutting part's shape and the absence of cracking. An increase in the strength of carbide inserts for turning tools treated by the pulsed magnetic field was due to stabilizing the stresses balance in the cobalt phase. This effect also prevents the spread of destructive cracks and improves material strength.

### 3.3. Experimentally Determining the Thickness of the Hardened Layer and the Surface Hardness after Nitriding

The microstructure of diffusion layers obtained by nitriding various steels in the container with a nitrogen-containing substance is similar to the diffusion layers following traditional gas nitriding (Figures 1 and 2).

An increase in the temperature leads to an increase in the thickness of the nitride layer. The curves' nature is the same for all investigated steels. However, the layer's depth strongly depends on the steel composition. For all the studied alloy steels, an increase in the nitriding temperature from 450 °C to 700 °C leads to an increase in the diffusion layer thickness from 0.15 mm to 0.28 mm (at the temperature of 450 °C), and from 0.45 to 0.70 mm (at the temperature of 700 °C). As is well known, the traditional long gas nitriding for 40–100 h layer thickness decreases with an increase in carbon and alloying elements contents in the steel. This fact is due to the coagulation, growth, and spheroidization of carbides and nitrides in alloying elements for such a continuous chemical-heat treatment. Notably, this process does not occur at short-term nitriding (up to 5 h). During this time, the alloying elements' carbides and nitrides do not have time to grow to full sizes which inhibits further nitrogen diffusion into the metal. The presence of carbon and alloying elements [42] increases lattice defects and accelerates diffusion [43].

Changes in the time of nitriding from 1 to 5 h lead to an increase in the depth of the diffusion layer from high-speed tool steel W6Mo5 from 127 μm to 210 μm. However, increased exposure of up to 5 h to obtain a deeper layer is accompanied by increased brittleness. In this regard, notably, the brittleness of the layer is an essential issue. A fragile layer at work friction can peel; small particles that broke off, unable to continue to play the role of abrasive, contributing to the rapid wear. The great fragility of the layer and

other properties of parts also decreases. Therefore, the optimal selectable exposure time is 1 h. The diffusion layer's total thickness is 127 μm, and the nitride phase is formed with a thickness of 35–40 μm. This phase relationship provides a significant performance of the cutting tool.

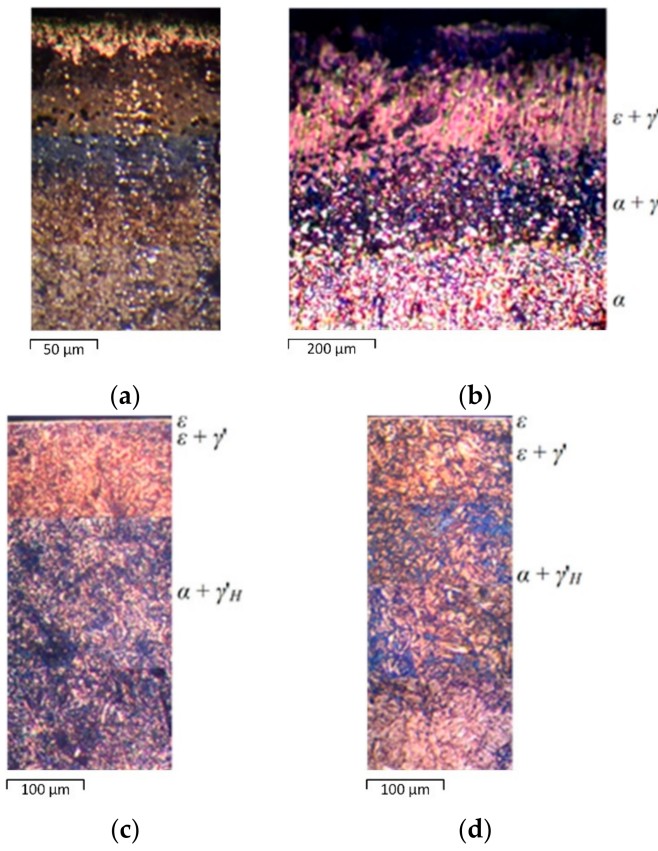

**Figure 1.** Microstructure of steels W6Mo5 (**a**), 105MnCrW11 (**b**), 40Cr (**c**), and 18Cr2Ni4Mo (**d**) after nitriding at 550 °C for 5 h.

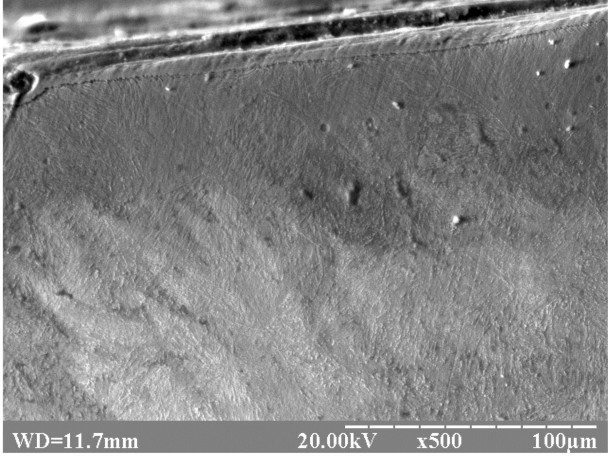

**Figure 2.** Microstructure of steel 38Cr2MoAl after nitriding at 550 °C for 5 h.

Microhardness distribution on the depth of steels' diffusion layers after nitriding for 5 h at different temperatures showed that all steels' high hardness corresponds to 450–500 °C. However, the total thickness of the nitrided layer at such a temperature is insufficient. Increasing the temperature from 450 °C to 700 °C significantly reduces the surface hardness

and increases the layer's thickness. Given this performance, the optimization is the ratio of the surface hardness to the thickness of the nitride layer [44], the optimum nitriding temperature of 550 °C was chosen [45].

A study of microhardness changes from the surface to the core shows that the nitrided layer's hardness depends on the carbon content and alloying elements. Greater content indicates higher hardness due to the additional formation of nitrides and carbides of alloying elements. The investigation of the nitriding duration effect on the diffusion layer depth shows that an increase in nitriding time up to 5 h leads to a significant increase in the layer thickness. However, with further aging, the growth rate of the layer thickness is much slower.

After nitriding the samples in the container, X-ray radiation analysis of the studied steels was conducted. The diffraction pattern of the surface shows the presence of nitrides $\xi$-Fe$_2$N, $\varepsilon$-Fe$_3$N–Fe$_2$N, $\gamma'$-Fe$_4$N, Fe$_3$N that $\alpha$-Fe. The highest intensity corresponds to nitrides Fe$_3$N–Fe$_2$N, Fe$_4$N.

In alloy steels, nitrides and carbides of alloying elements are fixed. For steel 40Cr, there are CrN, Mn$_4$N, and Fe$_3$C; for steel 18Cr2Ni4Mo—CrN, MoN, Mo$_2$N, Mn$_4$N, Fe$_3$C, and Cr$_7$C$_3$. For the case of the high-speed tool steel W6Mo5, a diffraction pattern showed the presence of nitrides of alloying elements Cr$_2$N, W$_2$N, Mo$_2$N, and VN, carbides Fe$_3$C, WC, W$_2$C, MoC, Mo$_2$C, VC, V$_2$C, Cr$_3$C$_2$, Cr$_7$C$_3$, and Cr$_{23}$C$_6$, as well as a solid solution of nitrogen in $\alpha$-Fe (Figure 3).

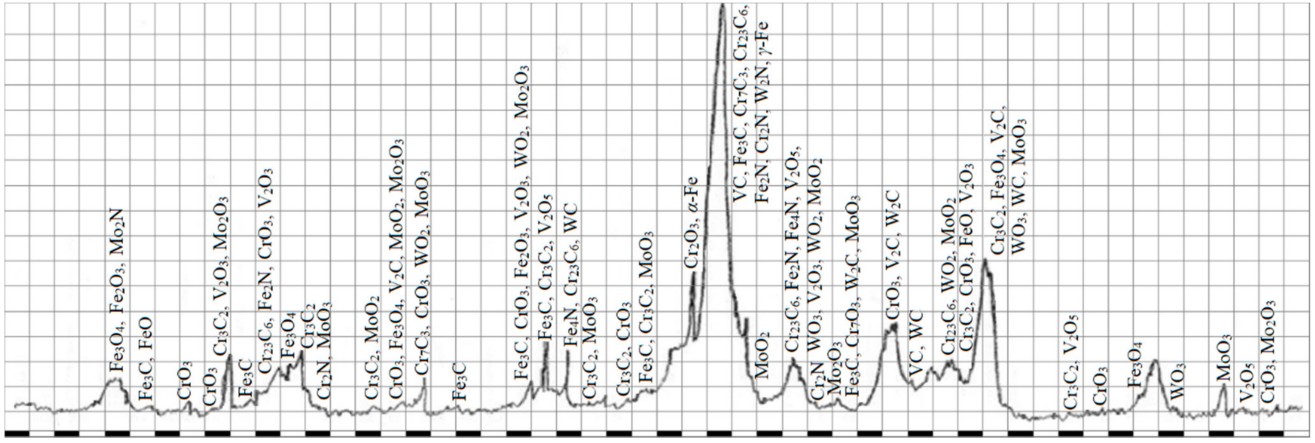

**Figure 3.** The diffraction pattern of the high-speed tool steel W6Mo5 on the surface after nitriding at a temperature of 550 °C for 1 h.

Thus, nitriding using a nitrogen-containing substance at the developed optimal mode leads to a 10-fold acceleration of the process, primarily due to the increase in the nitrogen atomic number.

### 3.4. The Thickness of the Hardened Layer and the Surface Hardness after Boriding Steel Products

The fundamentals for developing hardening tools and structural materials allow us to intensify the saturation of atomic boron products' and improve their manufacturability. The proposed coating consists of boriding in the nano-dispersed substance with the activators. A significant advantage is that this allows the boriding process to be carried out in a conventional oxidizing atmosphere without special equipment, sealing, and the use of protective atmospheres.

It is known that the temperature of the boriding can be 800–1200 °C, depending on steel grades and properties that must be obtained from the boride layers [46,47]. Up to 1100 °C, the needle-shaped borides are formed with a hardness of 20–23 GPa (in a transition zone from them to the core metal). At 1100–1200 °C on the steel surface, a eutectic structure is formed with a lower hardness of 13–16 GPa. For most structural steels, the boriding temperature is chosen close to the quenching temperature to combine these two treatments.

Increasing the temperature from 800 to 1000 °C increases the diffusion layer thickness. However, this growth leads to the grains' growth, causing the steel's diffusion layer's fragility. Below 800 °C, boriding is impracticable due to relatively slow diffusion. The borated layer formation on the first steel from the surface deep into the metal germinates a separate needle-shaped crystal of borides $Fe_2B$. Gradually, these crystals coalesce into a continuous layer. Upon further saturation with boron, another layer's surface of borides FeB is formed. Below, the borated layer "selection of boron cementite" as phase composition $Fe_3(B,C)$ is formed due to the carbon displacement from the borated surface layer.

The character structure for all steels except high-speed steels was investigated. The corresponding temperatures coincided with the hardening temperature for each steel because the combination of boriding and annealing is the most rational. The quenching temperature for high-speed steels is 1250–1280 °C, and combining boriding with hardening is impossible because of the melting of the surface due to the formation of the eutectic structure. This effect leads to a significant reduction in hardness of up to 13–15 GPa. Therefore, boriding these steels was carried out after hardening. The boriding temperature is 1000 °C, allowing the surface layer of the two borides FeB and $Fe_2B$ with high surface hardness.

The nature of the steel structure after boriding for 2 h is shown in Figures 4 and 5.

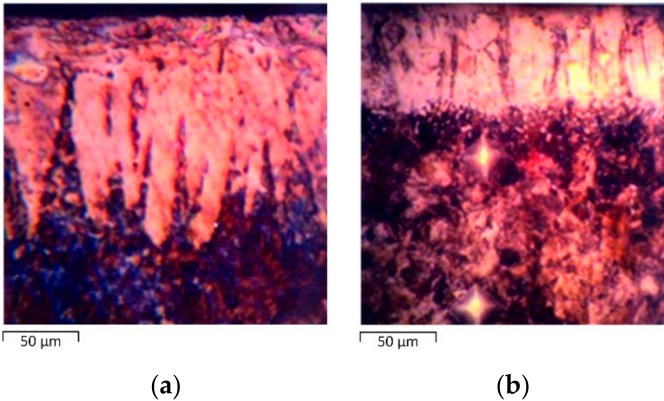

(**a**)　　　　　　　　　　　　(**b**)

**Figure 4.** The microstructures of the boride layers on steel 40Cr (**a**) and the bearing steel Cr15 (**b**) after boriding at a temperature of 850 °C for 2 h.

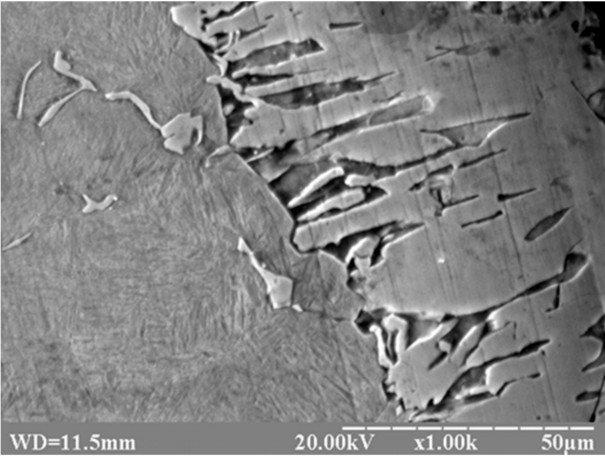

**Figure 5.** The microstructures of the boride layers on steel 38Cr2MoAl 40Cr after boriding at 950 °C for 2 h.

Their comparison follows that, despite the same temperature (850 °C) and time (2 h), the layers differ in thickness and have some features. Thus, in bearing steel Cr15 in the front, saturation is flat, with no detectable needle structure. Metallographic studies have

shown that steel 40Cr typically has wide wedges, while those in steel 18Cr2Ni4Mo are much thinner.

There are differences in the phase composition. On the diffraction pattern of bearing steel Cr15 (Figure 6), except for the borides FeB and $Fe_2B$, there is a chromium boride ($Cr_2B$) and a special carbide $Cr_7C_3$. The alloy steel 18Cr2Ni4Mo comprises the borides of chromium CrB and $Cr_2B$, borides of molybdenum $MoB_2$ and $Mo_2B$, and a cementite phase.

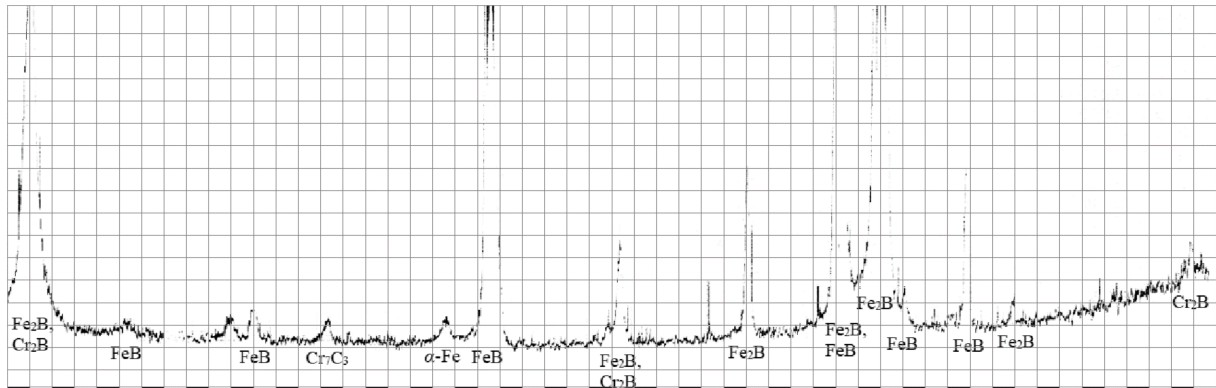

**Figure 6.** The diffraction pattern of the bearing steel Cr15 on the surface after boriding at a temperature of 850 °C for 2 h.

Consequently, in bearing steel Cr15 in the front, the saturation is flat, with no detectable needle-like structure. Metallographic studies show that steel 40Cr typically has wide wedges, which are much thinner in steel 18Cr2Ni4Mo. There are differences in the phase composition. On the diffraction pattern of bearing steel Cr15, in addition to borides FeB and $Fe_2B$, there is a boride of chromium ($Cr_2B$) and a special carbide $Cr_7C_3$. The alloy steel 18Cr2Ni4Mo comprises the borides of chromium CrB and $Cr_2B$, borides of molybdenum $MoB_2$ and $Mo_2B$, and a cementite phase.

In high-speed steels (W18, W6Mo5), the worsening of wedges borides is significantly reduced before their rounding (Figure 7). This fact is associated with the diffusion inhibition of boron alloying elements. These steels are characterized by a solid boride layer and separate rounded sections of borides under it. Layer-by-layer, X-ray phase analysis confirmed the presence of borides FeB, and $Fe_2B$, borides of alloying elements $Cr_2B$, CrB, $Cr_3B_4$, $Mo_2B$, $Mo_2B_5$, $MoB_2$, $W_2B$, and $W_2B_5$, and carbides of $Cr_7C_3$, $B_4C$ $Fe_3C$, VC, and WC in the diffusion layer.

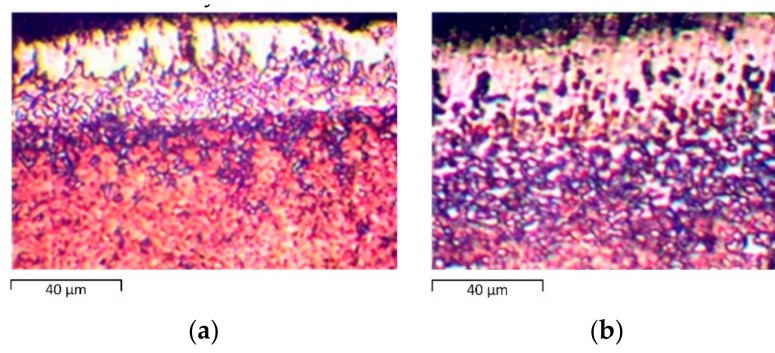

(**a**)            (**b**)

**Figure 7.** The microstructures of the boride layers on steel high-speed tool steel W6Mo5 (**a**) and high-speed tool steel W18 (**b**) after boriding at the temperature of 1000 °C for 2 h.

According to experimental data, the dependence of the thickness of the boride length of boriding for different steels (Table 1) shows that the thickness of the boride layer for all steels increases with the duration of the process. The rate of growth is highly dependent on the composition of the steel.

**Table 1.** The effect of the duration of the boriding of steel products on the change of the boride layer.

| Steel Grade | Boriding Temperature, °C | The Layer Thickness of Borides, µm, for the Different Boriding Duration | | | | | | Surface Hardness (Microhardness), GPa |
|---|---|---|---|---|---|---|---|---|
| | | 15 min | 30 min | 45 min | 60 min | 90 min | 120 min | |
| Bearing steel Cr15 | 850 | 14 | 31 | 44 | 54 | 66 | 73 | 21.4 |
| W18 | 1000 | 8 | 10 | 12 | 16 | 29 | 49 | 23 |
| W6Mo5 | 1000 | 3 | 4 | 6 | 10 | 22 | 38 | 23 |

The surface layer's hardness also depends on the steel composition. However, to a much lesser extent, the hardness of borides (FeB, $Fe_2B$), which are the main phases of the surface layers in the carbon content and alloying elements, slightly depends on the steel composition. Microhardness confirms the presence of two borides FeB with the hardness of 20–23 GPa, and $Fe_2B$—with the hardness of 15–18 GPa. Structural carbon steels [48] are characterized by a smooth microhardness distribution in the transition zone than in the tool. The surface hardness is in the range of 20–22 GPa for structural carbon steel. Additionally, the surface hardness is up to 21–23 GPa for cutting alloy steel.

Thus, in order to simplify the technological process of hardening treatment of all steels (except high-speed steels), it is proposed to combine the boriding operation with the hardening operation. For these, it is advisable to carry out boriding at a temperature of 1000 °C after hardening. The duration of boriding is selected under the requirements of the parts. The corresponding time should not be less than 30 min for structural steels, 45 min for carbon tool steel, and 120 min for high-alloyed steels.

The evaluation of abrasive wear resistance of steels without CHT and after nitriding and boriding on the developed technologies showed the samples' linear wear. The durability $I_{wr}$ of wear resistance was evaluated using the wear rate $I_r$ ($I_{wr} = 1/I_r$, where $I_r$—the ratio of the wear $I$ along with the path $L$, which has been fraying $I_r = I/L$).

Studies have shown the linear dependence of the wear from the friction path for all steels for both nitrided and borated conditions. After boriding, wear is significantly less than it is following nitriding [49,50]. On the probationary path of 200 m, the amount of wear after boriding 2.1–3.6 times (depending on steel grades) is less than that following nitriding. This fact is primarily due to the significantly higher hardness of the boride layers. For nitrided or substantially horizontal samples of curves, durability does not depend on friction (steel 40Cr, and bearing steel Cr15) or is slightly reduced at the initial testing stage (high-speed tool steel W6Mo5). After boriding all steels, wear resistance considerably increases at the initial stage (up to 120 m of the way of friction, for high-speed tool steel W6Mo5—80 m), and then it slowly changes.

Experimental data showed that the cutting tool equipped with inserts from WC92Co8 and WC85TiC6Co9 is more resistant to the untreated steel WC92Co8 by 183% and WC85TiC6Co9—more than 200%. It is possible to increase wear resistance after nitriding up to 120–177% and boriding and up to 180–340% depending on alloy steel.

The scientific novelty of this research, among other works, is as follows. For the first time, a decrease in the coefficient of variation of microhardness, stability and an increase in the stability of the physical and mechanical properties of carbide cutting tools after processing with a pulsed magnetic field was theoretically determined and experimentally established. Additionally, based on the analysis of the patterns of structure formation of diffusion layers, it was found that the use of nanocrystalline nitrogen-containing powder allows us to obtain the necessary ratio of the surface layer hardness and its depth distribution, which allows us to increase the wear resistance of the machine parts and tools, as well as accelerate the process of the nitriding of alloy steels in comparison with traditional gas nitriding by ten times without the use of special equipment. Finally, it was found that the new proposed nanodispersed boron-containing powder intensifies diffusion processes, which accelerates boriding during furnace heating by 2–3-old compared to existing technologies while significantly increasing the wear resistance of machine parts and tools.

## 4. Discussion

The results obtained by determining the hardened layer's thickness and the surface hardness after the MPT products of steel Cr15 allowed us to study the effects of MPT on the reliability of cutting tools made from hard alloy WC85TiC6Co9 and to determine the thickness of the hardened layer. The surface hardness after nitriding and boriding indicates that the proposed methods for the surface hardening of alloy steels and hard alloys significantly increase the hardness of surface layers. This fact can significantly increase the service life of tools and machine parts.

After MPT of steel Cr15, the microstructure of the alloy is improved. This effect increases the surface microhardness by 40–50% compared with baseline (2.4 GPa). The depth of the hardened layer is 0.08–0.10 mm.

After the magnetic pulse treatment, the surface microhardness distribution of the investigated hard alloys WC92Co8 and WC85TiC6Co9 relative to the untreated one increased to 17 GPa. The variation coefficient for the microhardness decreased from 0.13 to 0.06, and stability decreased from 0.48 to 0.27. The change in lattice parameters of Co and TiC after processing the alloy shows a reduction in the lattice parameter and distortion. It also confirms the deformation of the cobalt phase solid alloy, with an increase in the strength of the hard alloy. Stability of the cutting tool equipped with the inserts from WC92Co8 and WC85TiC6Co9 processed by the MPT method, relatively unprocessed, increases by 1.8–2.0 times in the absence of cracking. Obtaining such data will increase the wear resistance of the tool by 2.5–3.0 times.

It should be noted that an increase in the stability of the cutting properties of the hard alloy is associated with recrystallization, the release of the dispersed phase, and the homogenization of the cobalt phase under the influence of the pulsed magnetic field. The use of MPT increases the stresses transition intensity in the cobalt phase from a tensile to compressive one. It also increases the strength and stability of the tool's cutting properties.

However, the effectiveness of the developed nitriding and boriding technologies significantly reduces the duration of treatments with sufficient hardened layers. The microstructures of hardened layers confirm the classical diffusion of layers with distinct nitride, boride, and transition zones. On the other hand, the result removes the contradiction between the need to achieve high performance and productivity. The latter relates to the problem of improving the composition of the saturating medium and the technological conditions of hardening treatments. That is why the proposed methods of surface hardening are promising.

Remarkably, the proposed method of nitriding based on the saturation of the surface of steel with atomic nitrogen from the dispersed medium in a closed container does not consider such an important fact as the processing of large-sized products. This result is due to the size of the container and the amount of saturated mixture. An increase in the size of the container can change the pressure in the inhomogeneity of the hardened layer. Therefore, this fact indicates the disadvantage of the proposed method. The direction of further research associated with its elimination should focus on developing the mixture's saturation capacity for large-sized products. This adaptation should make it possible to obtain a uniform hardened layer of the part regardless of size.

## 5. Conclusions

The proposed method of surface hardening based on the MPT in its sufficient simplicity which allows to increase the surface microhardness of steel Cr15 by 40–50% was compared with baseline (2.4 GPa). As a result, the depth of the hardened layer of 0.08–0.10 mm was obtained. It has been shown that treatment with magnetic field pulses of tool carbide reduces heterogeneity and residual stress levels and eliminates structural defects. These facts positively affect the tool's further maintenance. A decrease in the variation coefficient for microhardness (from 0.13 to 0.06) and stability (from 0.48 to 0.27) are essential factors for the optimal choice of the MPT method. This effect indicates an increase in the stability of hard alloys' physical and mechanical properties. A distinctive feature of the cutting

tool equipped with inserts from WC92Co8 and WC85TiC6Co9 alloys is the high resistance relative to the untreated WC92Co8 higher by 183%, and WC85TiC6Co9—more than 200%. The proposed method improves the carbide cutting tool's reliability and stabilizes its physical and mechanical properties.

For obtaining a hardened surface layer of alloyed steels, applying the nitriding method in the container using nanocrystalline powder is effective. Obtaining the depth of the nitrided layer to 0.3–0.5 mm (depending on the steel composition) with an increase in surface microhardness to 12.7 GPa allows us to accelerate the processing by 10-fold compared to traditional gas nitriding. This ratio of the surface layer's hardness and its distribution on the steel depth increases the wear resistance of machine parts and tools by 120–177%, depending on the lightness. A distinctive feature of this method is the need for long-term aging in the furnace products, which does not lead to the decoupling of the steels' matrix. This feature of the method simplifies the process, which significantly reduces energy costs. This fact becomes possible due to the proposed method's nano-dispersed powder as a saturating medium. Confirmation of this fact can be found in the study of microstructures of the diffusion layer, which showed that the resulting nitrided layer is similar to the layers obtained in other nitriding types.

The proposed method of boriding can be implemented to strengthen the products and tools of complex shape, especially if they are after hardening treatment are not subjected to heat treatment, as the corrosion and deformation at these methods are minimal. For making this, it is necessary to use nano-dispersed boron-containing powder, which provides a surface hardness of alloyed steels of up to 21–23 GPa. Due to nano-dispersed powder, the boriding process is accelerated by 2–3 times compared to existing technologies. Overall, this hardening characteristic allows us to increase the wear resistance of steels by 180–340%.

**Author Contributions:** Conceptualization, K.K.; methodology, K.K., V.K. (Viktoriia Kostyk) and M.S.; software, V.I. and I.K.; validation, I.P., V.K. (Viktor Kovalov) and K.K.; formal analysis, I.K. and M.S.; investigation, K.K.; resources, V.K. (Viktoriia Kostyk) and I.P.; data curation, V.K. (Viktoriia Kostyk) and V.I.; writing—original draft preparation, K.K.; writing—review and editing, V.I. and I.P.; visualization, V.K. (Viktor Kovalov); supervision, V.I. and M.S.; project administration, I.K.; funding acquisition, I.K. All authors have read and agreed to the published version of the manuscript.

**Funding:** This research was funded by the Slovak Research and Development Agency under contract No. APVV-16-0283. Project title: Research and development of multi-criteria diagnosis of production machinery and equipment based on the implementation of artificial intelligence methods.

**Institutional Review Board Statement:** Not applicable.

**Informed Consent Statement:** Not applicable.

**Data Availability Statement:** Not applicable.

**Acknowledgments:** Some of the results were also obtained within the project "Fulfillment of tasks of the perspective plan of development of a scientific direction "Technical sciences" Sumy State University" funded by the Ministry of Education and Science of Ukraine (State reg. no. 0121U112684). The research was partially supported by the Research and Educational Center for Industrial Engineering (Sumy State University) and International Association for Technological Development and Innovations.

**Conflicts of Interest:** The authors declare no conflict of interest.

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
