# Peer review of "Impact of Magnetic-Pulse and Chemical-Thermal Treatment on Alloyed Steels’ Surface Layer"

_applsci, doi:10.3390/app12010469_

Round 1

Reviewer 1 Report

English needs correction in the article. Different grammar, style and clarity problems were noticed in the text. Please find following examples (not all errors are listed below).

  • Line 82/83 - sentence: “Mainly, they are used for hardening wide-range machine parts” should be: Mainly, they are used for hardening wide-range of machine parts.
  • Line 211 - sentence: “Installation for metals’ magnetic-pulse treatment is a pulse current generator (PCG)” – in my opinion the apostrophe is in wrong place or it is unnecessary (depending of what authors had in mind) – same possible mistakes in lines: 228, 315 etc. In general I think that use of apostrophes may not be correct and should be double checked before publishing.
  • Line: 309 - sentence: “Surveillance of the cracks showed that destructive cracks spread in alloys of WC92Co8 and the WC85TiC6Co9, mainly in the cobalt phasen WC85TiC6Co9, where destructive crack spreads mainly by phase (Ti,W)C, the cobalt component can inhibit destructive crack. Phase cobalt carbide is a solid solution of W and C in a cubic.” Is really hard to understand correctly. It should be rewritten and corrected.
  • Line 312 - sentence: “Phase cobalt carbide is a solid solution of W and C in a cubic” is somehow hard to understand. It should be rewritten like this: “Cobalt-carbide phase has a cubic solid solution structure of W and C.
  • Line 325 – sentence: “Application vibro-abrasive treatment before exposure to the magnetic field enhances the intensity of the transition stresses in the cobalt phase from tensile to compressive.” Is missing “of” after ”Application”.
  • Line 388 – sentence:After nitriding, the samples in the container, X-ray radiation analysis of the studied steels was conducted” is hard to understand – maybe coma after nitriding is unnecessary ? Please correct the sentence to be more clear.
  • Line 407 – sentence: “that this allows the paste boriding process in a conventional oxidizing atmosphere” – to paste the process ?.
  • Line 451 – sentence: “with no detectable needle structure” – there are no such thing as needle structure – do you mean “columnar grains” ?
  • Table 1 – Description of table needs to be rewritten – poor English. Style of the table is a little bit confusing (it would be much better if boring duration time (15, 30, 45 min …. were more clearly divided from results of measurements. Also within the table sentence: “The layer thickness of borides, μm, for the different boring duration” boriding instead of boring ?. These applies also for “Boring” temperature” which should be “Boriding temperature” Also add hardness measurement method (not only Pa).
  • Line 481 – sentence: “Thus, to simplify the technological process is proposed to combine boriding quenching all steels except high-speed.” Is hard to understand.
  • Line 482 – sentence: For them, it is helpful to boriding at a temperature of 1,000 °C after quenching.” Should be “For them, it is helpful to boride at a temperature of 1,000 °C after quenching.” ? I am not 100% sure about the correct conjugation here.
  • Line 491 – sentence: “After boriding, wear significantly less than after nitriding” is missing the “is” after “wear”.
  • Line 504 – sentences: “The scientific novelty of the research, among other works, is as follows. This article has the scientific novelty among other works as follows.” Both sentences means the same.
  • Line 562 – sentence: “The proposed method of surface hardening based on the MPT in its sufficient simplicity to increase surface microhardness of steel Cr15 by 40–50 %” is lacking of i.e. “allows” after “simplicity

Other problems found:

  • Line 33 - no unit for resistance values (also add type of the resistance),
  • Line 210 - I do not understand and can’t find any information regarding the “9.120.00.00.000”. Is it an installation number from a national grant or something else ? Please specify. Same for line 233 with “PPMF RK-1”, same for line 263 with “MIM-8”microscope, and same for line 275 with “AP 274 40.613.20 R 43/82”. I was unable to look for all above mentioned devices and set ups. If there is a possibility please specify a producer and some basic parameters (measuring methods) for those devices that cannot be easily identified.
  • What was the scale for microhardness measurements. In the article authors only indicate the value of MPa which is not sufficient (also the method must be specified (i.e. HV5).
  • Line 335- What authors meant by „absence of plastic changes”? Is it deformation or strain due to plastic flow of the metal ?

General remarks on the article:

  • As I understood from the article also nitriding process was realised for analysed samples. Why in the article there are no results (in tabularised form) of hardness measurements and thickness for nitride layer. Also research results for PMF surface treatment in form of surface hardness should be presented as well (in another table). It would be much more understandable if those research results were shown in that way and compared to each other.
  • Regarding to table 1 – The surface hardness was measured after what time of boriding process? Is it an average value (from how many individual measurements) from all measurements or only the value at the end of the process after 120 min. It would be much more beneficial for the article to show results of hardness after every time of boriding process (and after nitriding and PMF). Without those results it is somehow hard to understand results of performed research and hard to draw your own conclusions.

Author Response

The authors appreciate the time and comments devoted to the reviewing manuscript.

Reviewer 2 Report

Remarks:

Abstract (line 28).

1. The authors summarize their results: after magneto-pulse treatment, the hardening depth surface hardening 0.08…0.1 mm, after nitriding – 0.3…0.5 mm. However, after boration, the hardening depth is not represented (line 34).

The authors report that after boration, wear resistance increases by 180-340%, after nitriding – by 120-177%. However, after magneto-pulse treatment, the magnitude of the increase in wear resistance is not presented (line 40).

Introduction (line 44).

2. The authors write (line 77-80) "The surface hardening methods lead to hardening the product’s surface layer due to the following mechanisms and combinations: the substructure, solid solution, polycrystalline, and multi-phase hardening [4]". However, the article [4] is devoted to "Development of an iron-based alloy with a high degree of shape recovery" (line 623).

3. The authors write (line. 77-80) "For making deformations, hardening includes blasting, magnetic-pulse processing [6]" However, the article [6] is devoted to "An investigation of the temperature distribution of a thin steel strip during the quenching step of a hardening process" (line 627).

Materials and method (line 193).

4. It is better for the authors to formulate their goals and objectives (line 194) at the end of the "Introduction" section.

5. Why do the authors describe the time interval of nitriding in hours 1-7 hours (line 257), and boration – in minutes 15-120 min. (line 262)?

Results (line 277).

6. The average value of microhardness for samples before treatment was 2.4 MPa (line 280). This is probably a mistake, must be 2.4 GPa?

7. The absence of a scale segment in the photographs (Fig.1 (line 346), Fig.4 (line 440), Fig.7 (line 465)) makes it difficult to carry out a quantitative assessment of the structure.

8. Absence of a diffraction pattern from the steel surface before nitriding and boration (Fig.3 (line 398), Fig. 6 (line 448)), does not allow the analysis of structural-phase transformations during chemical - thermal treatment.

9. The entry in the table 1 (line 471) "The layer thickness of borides, µm, for the different boring duration, min" should be divided.

Author Response

(The authors gave the same response as above.)

Round 2

Reviewer 1 Report

Thank for all your corrections and explanations.

I have only one more remark regarding the hardness measurement method which was still not revealed (I understand it was made with the PMT-3 tester but what was the indenter shape ?). Was it Vickers method? If so please specify this somewhere in the article and also please provide the load used to make above mentioned measurements in the text itself (from your response letter I understand it was both 5 and 10g ? - are there any significant differences in the values obtained ? If not please clarify this in the text).

Just for example, the results from hardness measurements should look like in the following article.

https://www.researchgate.net/publication/262573356_Investigation_of_Tribological_Properties_and_Characterization_of_Borided_AISI_420_and_AISI_5120_Steels/figures?lo=1&utm_source=google&utm_medium=organic

Also regarding your response: “The surface hardness does not depend on the duration of boriding treatment, this is a well-known fact. This is due to the formation of boride FeB on the surface of the steel, the hardness of which is 20–23 GPa. Therefore, it does not make sense to write the same numbers for all time modes of boriding.”

Yest it is true, but nevertheless it should me mentioned when you measured the hardness, how many indentations you made, what was the average value etc. Giving only one value without any information about the spread of the results is not sufficient. As you probably know microhardness measurements can give difference results even at the same sample depending from the structure of the sample, quality of surface layer etc. Please provide more information about the nature of hardness measurements (i.e. how many indentations were made on sample and what was the average value etc.)

Best Regards

Author Response

Thank you very much for your interest in our work and the desire to improve our article to make it more understandable for the entire world community!
